# Pax-5 Protein Expression Is Regulated by Transcriptional 3′UTR Editing

**DOI:** 10.3390/cells11010076

**Published:** 2021-12-28

**Authors:** Annie-Pier Beauregard, Brandon Hannay, Ehsan Gharib, Nicolas Crapoulet, Nicholas Finn, Roxann Guerrette, Amélie Ouellet, Gilles A. Robichaud

**Affiliations:** 1Department of Chemistry and Biochemistry, Université de Moncton, Moncton, NB E1A 3E9, Canada; annie_pier_beauregard@hotmail.com (A.-P.B.); brandonlhannay@gmail.com (B.H.); ehsan.gharib@umoncton.ca (E.G.); Nicolas.Crapoulet@vitalitenb.ca (N.C.); roxannguerrette@gmail.com (R.G.); amelieouellet5@gmail.com (A.O.); 2Atlantic Cancer Research Institute, Moncton, NB E1C 8X3, Canada; 3Dr. Georges-L-Dumont University Hospital Centre, Moncton, NB E1C 8X3, Canada; Nicholas.Finn@vitalitenb.ca

**Keywords:** *Pax-5*, alternative polyadenylation, 3′UTR editing, B-cell, cancer, alternative splicing, microRNA

## Abstract

The *Pax-5* gene encodes a transcription factor that is essential for B-cell commitment and maturation. However, *Pax-5* deregulation is associated with various cancer lesions, notably hematopoietic cancers. Mechanistically, studies have characterized genetic alterations within the *Pax*-5 locus that result in either dominant oncogenic function or haploinsufficiency-inducing mutations leading to oncogenesis. Apart from these mutations, some examples of aberrant *Pax-5* expression cannot be associated with genetic alterations. In the present study, we set out to elucidate potential alterations in post-transcriptional regulation of *Pax-5* expression and establish that *Pax-5* transcript editing represents an important means to aberrant expression. Upon the profiling of *Pax-5* mRNA in leukemic cells, we found that the 3′end of the *Pax-5* transcript is submitted to alternative polyadenylation (APA) and alternative splicing events. Using rapid amplification of cDNA ends (3′RACE) from polysomal fractions, we found that *Pax-5* 3′ untranslated region (UTR) shortening correlates with increased ribosomal occupancy for translation. These observations were also validated using reporter gene assays with truncated 3′UTR regions cloned downstream of a luciferase gene. We also showed that *Pax-5* 3′UTR editing has direct repercussions on regulatory elements such as miRNAs, which in turn impact Pax-5 protein expression. More importantly, we found that advanced staging of various hematopoietic cancer lesions relates to shorter *Pax-5* 3′UTRs. Altogether, our findings identify novel molecular mechanisms that account for aberrant expression and function of the *Pax-5* oncogene in cancer cells. These findings also present new avenues for strategic intervention in *Pax-5*-mediated cancers.

## 1. Introduction

Research has enabled the elucidation of multiple regulatory pathways in aberrant gene expression leading to disease, thus presenting new therapeutic targets and strategies. Adding complexity to these intricate networks are non-coding RNAs and the myriad of post-transcriptional modifications of mRNA targets. In fact, mRNA editing is biologically essential for the expansion of gene products and functional diversity. Studies estimate that up to 95% of human gene transcripts are subjected to alternative splicing events [1,2,3]. Reciprocally, reports also show that aberrant alternative splicing is frequently associated with the development of cancer [4].

Accordingly, this is also the case for the *Pax-5* gene, which encodes a transcription factor essential for B-cell lineage commitment and maturation [5,6,7]. Given its pivotal regulatory function in B-cell development, *Pax-5* is also involved with the onset and progression of various hematological cancers and some carcinomas (reviewed in [8]). We and others have previously shown that the *Pax-5* transcript is submitted to multiple post-transcriptional editing processes [9,10] and alternative promoter regions [11,12,13] to regulate protein expression. In addition, genetic studies have shown that the human *Pax-5* locus is involved in recurrent chromosomal rearrangements, which account for the aberrant expression of the *Pax-5* oncogene in cancer. For example, translocation of the *Pax-5* gene can result in chimeric transcription factors [14,15] or position the *Pax-5* coding region under transcriptional control of the potent immunoglobulin heavy chain (*IgH*) gene promoter [16]. Although genomic instability is the root of aberrant *Pax-5* expression in most cancer malignancies, some studies have ruled out genetic alterations involving *Pax-5* as a causal link to deregulated Pax-5 protein expression in cancer cells, thus suggesting an alternate mechanism for *Pax-5* ectopic expression [10,17]. This lack of understanding may also be due to the paucity of studies on the post-transcriptional regulation of *Pax-5* products.

We have previously reported that *Pax-5* expression and function are modulated by key micro-RNAs (miRNAs) in various types of cancer cells [18,19,20]. Consequently, the transcripts’ untranslated regions (UTR), notably at the 3′end, plays a pivotal role in transcript stability and translation efficiency [21,22,23]. Furthermore, alteration and editing of mRNA 3′UTRs, which harbor multiple binding sites for translational regulatory elements, lead to concomitant changes in mRNA stability and expression. Interestingly, the structure and overall lengths of 3′UTRs are shown to vary widely in mammalian cells. This phenomenon is due to alternative polyadenylation (APA) signals that enable premature termination of transcription, which results in truncated mRNAs [24,25]. The use of APA motifs is observed in more than half of all human transcripts and represents an important means to evade translational control [24]. In fact, shortening of mRNA 3′UTRs is a prominent process observed in proliferating cells and oncogenes, leading to greater mRNA stability and translation [26,27].

In this study, we set out to characterize *Pax-5* 3′UTR variability and its repercussions on translational expression. We demonstrate that the *Pax-5* 3′UTR is significantly edited in cancer cells through the use of APA or alternative splicing events, depending on the tissue type. Although *Pax-5* 3′UTR editing events are prevalent in healthy B-lymphocytes, we associate *Pax-5* 3′UTR shortening with an increase in oncogenic translation frequency, which can be associated with disease progression of B-cell cancers. Our findings present the first characterization of *Pax-5* 3′UTR editing and provide new insight to avenues in aberrant expression of the *Pax-5* oncogene.

## 2. Materials and Methods

### 2.1. Cell Models and Culturing Conditions

MCF7 (mammary ductal carcinoma, HTB-22); MCF10A (mammary epithelial CRL-10317); Raji (lymphoblast B-cell, CCL-86), and REH (Acute Lymphocytic Leukemia, CRL-8286) were obtained from American Type Culture Collection (ATCC, Rockville, MD, USA). Nalm6 (B-cell precursor leukemia, ACC-128) was obtained from the Leibniz Institute DSMZ culture collection (Braunschweig, Germany). MCF7 cells were cultured in DMEM medium supplemented with 10% fetal bovine serum (FBS) and L-glutamine (2 mM). MCF10A cells were maintained in DMEM/F12 medium supplemented with 5% FBS, L-glutamine (2 mM), sodium pyruvate (1 mM), bovine insulin (10 μg/mL), EGF (20 ng/mL), cholera toxin (100 ng/mL), and hydrocortisone (500 ng/mL) (ThermoFisher, Burlington, ON, Canada). B-cells were maintained in RPMI 1640 medium supplemented with 10% FBS and L-glutamine (2 mM). Healthy donor and patient samples were obtained from the G.L.-Dumont University Hospital Centre (GLDHC, Moncton, NB, Canada) biobank repository and conducted in accordance with the Declaration of Helsinki of 1975 (revised 2013, https://www.wma.net/what-we-do/medical-ethics/declaration-of-helsinki/) (accessed on 1 February 2020). Ethics approval was provided by the GLDHC (project: CER7-3-17 Ver. 5) and the Université de Moncton (Moncton, NB, Canada) (project: 1920-016). Patient samples consisted of peripheral blood mononuclear cells (PBMCs) isolated by gradient centrifugations with Ficoll Paque (GE Life Sciences, Mississauga, ON, Canada). Primary CD19+ B-lymphocytes were purified from Ficoll density gradients of PBMC using a magnetic B-cell isolation kit according to the manufacturer’s instructions to isolate untouched B-cells by negative selection (Miltenyi Biotec, Somerville, MA, USA).

### 2.2. PCR, Cloning, and Quantitative RT-PCR

Reverse transcriptions were performed using SuperScript III reverse transcriptase (ThermoFisher) on RNA retrieved from cells using TRIzol^®^ reagent (ThermoFisher) according to manufacturer’s instructions and previously described [28]. Levels of gene expression were verified by quantitative RT-PCR (qRT-PCR) in reaction mixtures composed of 12.5 μL of SYBR^®^ Green 2X FastMix (QuantaBio, Beverly, MA, USA), 2.5 μL of each forward and reverse primer (3 μM) and 2 μL of cDNA. For miRNA analysis, cDNA was generated from 5 ng of miRNA-enriched total RNA using TaqMan^®^ microRNA Reverse Transcription Kit (ThermoFisher). MicroRNA qRT-PCR assays were then performed using miRNA-specific TaqMan^®^ assays (ThermoFisher). Reactions were run in a real-time PCR CFX instrument (Bio-Rad Laboratories, Mississauga, ON, Canada). Comparative expression levels for mRNA or miRNAs were calculated using the ΔΔCt method [29], using the hypoxanthine ribosyltransferase (HPRT) or small nucleolar RNA48 non-coding transcript as a normalizing control, respectively.

The *Pax-5* 3′UTR regions were isolated by PCR amplification using *Pax-5* specific primers and cloned into either the pMiR-REPORT™ vector (ThermoFisher) or pGEM-T easy vector (Promega, Madison, WI, USA) for sequence and expression profiling. Rapid Amplification of cDNA Ends (RACE) was performed using the GeneRacer kit (ThermoFisher) as described by the manufacturer’s instructions with some modifications. Briefly, total RNA was extracted from cells and reverse transcribed using a provided modified oligo-dT adaptor primer. PCR reactions were then completed with a forward gene-specific primer and the provided GeneRacer 3′ adapter reverse primer. Nested PCR reactions were then performed using internal primer pairs specific to the *Pax-5* 3′UTR targeted region. All developed primers for specific PCR reactions are presented in Appendix A.

### 2.3. Reporter Gene Assays and Transfection

Three firefly luciferase-based reporter gene constructs (pMiR-REPORT, ThermoFisher) were designed with the luciferase gene located upstream from the various cloned regions of the *Pax-5* 3′UTR. Reporter constructs were transfected by electroporation (Nucleofector^®^, Lonza, Basel, Switzerland) into B-cell lines. Technically, 2 μg of DNA plasmid was transfected into 2 × 10^6^ cells using the Nucleofector^®^ II device (software version S4) and the Amaxa Cell Line Nucleofector Kit V according to the manufacturer’s instructions (Lonza). After a 48 h incubation period, cells were examined for luciferase expression using the Dual-Glo luciferase system (Promega) and a Synergy Microplate reader (Biotek, Winooski, VT, USA). In co-transfections (pMiR reporter constructs + anti-miRNAs), we used the Dharmafect DUO reagent (Horizon, Lafayette, IN, USA) where 2 × 10^6^ cells received 2 μg of reporter plasmid, 20 ng of Renilla luciferase plasmid (Promega), and 100 nM of miRNA inhibitors (anti-miR™ inhibitor, ThermoFisher) against specific miRNAs or scrambled non-targeting anti-miRs as negative controls. Transfection normalizations were performed either by qRT-PCR on a plasmid backbone gene (puromycin, located on pMiR-REPORT) or by co-transfection and detection of non-inducible Renilla luciferase (pRL, Promega).

### 2.4. Polysomal Fractionation and Profiling

Profiling of *Pax-5* transcripts in translation was performed with polyribosome extracts and fractionation using sucrose gradients (7–47% range), collection tubes, and absorbance monitoring as described previously [30]. Briefly, B-cells were pelleted and resuspended in RPMI 1640 media with 100 μg/mL of cycloheximide (CHX) (Sigma-Aldrich, Oakville, ON, Canada). After 15 min of incubation, cells were washed twice in PBS/CHX and lysed with a detergent buffer and pestle strokes in a Dounce homogenizer. The solution was centrifuged and added on top of a sucrose cushion (1 M sucrose in a polysome buffer) to concentrate the ribosomes. The polysomal fractions were collected using an ISCO Foxy Jr fractionator (Teledyne ISCO, Lincoln, NE, USA). The fractions were subjected to RT-PCR to establish differential *Pax-5* 3′UTR isoform quantification and translational activity using *Pax-5* specific primer pairs (Appendix A).

### 2.5. Data Mining and Processing

Illumina Hiseq 2000 RNA-seq data sets (level 3 per-gene RNA-seq v2 expression data) of diffuse large B-cell lymphoma (DLBCL) patients (*n* = 48) were retrieved from TCGA (https://cancergenome.nih.gov) (accessed on 10 April 2020). The dbGaP accession number to the specific version of the TCGA data sets is phs001444.v2.p1. For chronic lymphocytic leukemia (CLL) analysis, the RNA-seq profiles were downloaded from the Gene Expression Omnibus (GEO) data set GSE66117, including 47 CLL primary B-cells raw data [31]. For control, the GSE16921 data set containing the RNA-seq of 41 immortal B-cells [32] along with the GSE62246 [33,34] and GSE70830 including 2 and 5 normal B-cell RNA-seq profiles respectively, were considered.

FASTX-Toolkit (http://hannonlab.cshl.edu/fastx_toolkit/) (accessed on 10 April 2020) was used for quality assessment of the raw files. The error ratio was calculated by using the formula: Phred Quality (Q) score = −10 log_2_E (E = error rate). The homogeneity of sequences was measured based on the distribution of guanine–cytosine (GC) pairs. Adapters, reads with low quality, reads consisting of >50% N bases, or >20% N bases with the Q-score < 20 were excluded from the analysis. TopHat2 package version 2.1.1 [35] was applied for splice alignment of the filtered reads with the human reference genome (GRCh38.p12, Ensembl.org). The parameters were set as the default value.

The read counts of these data sets were estimated by RSEM package (RNA-seq by expectation maximization). To identify the protein-coding genes reads > 100 bp were analyzed with the Coding-Noncoding Index (CNI, version 2) based on 64 triplets of nucleotide algorithms [36], and additionally examined with the coding potential calculator-2 (CPC-2) webserver to distinguish noncoding transcripts from mRNAs based on the length and quality rate of the open reading frame of protein-coding transcripts [37]. Next, we analyzed the retrieved protein sequences with the Pfam-scan (version 1.3) database archive to evaluate their annotation [38] and also used the phylogenetic codon substitution frequency (phyloCSF, release 20121028) to distinguish the coding transcripts from noncoding alignments according to the evolutionary preservation of amino acids with known families of proteins [39]. Cuffdiff package (version 2.2.1, 2013) was used to estimate the expression score of the transcripts as RPKM = total exon reads/mapped reads in millions × exon length in kb (RPKM = reads per kilobase of transcript per million mapped reads). The false discovery rate (FDR) was chosen as 5, and the q-value (*p*-adjusted) was set as <0.05.

For the prediction of polyadenylation signals, we made use of a series of polyA signal prediction software: DNA polyA signal miner [40] (dnafsminer.bic.nus.edu.sg) (accessed on 17 February 2019); PolyApred [41] (http://crdd.osdd.net/raghava/polyapred/index.html) (accessed on 20 February 2019), and Softberry POLYAH [42] (http://www.softberry.com/berry.phtml?topic=polyah&group=programs&subgroup=promoter) (accessed on 20 February 2019). To examine the *Pax-5* transcript (notably the 3′UTR) for predicted splicing regulatory motifs and adenylate/uridylate-rich elements (AU-rich elements/AREs), we used the computational prediction software RNAreg 2.0 (http://regrna2.mbc.nctu.edu.tw) (accessed on 20 February 2019) [43]. For the validation of putative AU-rich motifs within the *Pax-5* 3′UTR, we also made use of ARED-Plus [44]. Finally, for the prediction of miRNA targeting motifs upon the *Pax-5* 3′UTR, we made use of a series of prediction software. First, microRNA.org (http://www.microrna.org/) (accessed on 20 March 2019) [45] used the old *Pax-5* reference sequence version number (NM_016734.1) from July 2013, which was characterized by a much shorter 3′UTR (2026 nucleotides (nts)). The score cut-off was set at −0.1 or lower for adequate prediction. A second tool, miRDB (http://mirdb.org/) (accessed on 20 March 2019) [46] used new reference gene versions (NCBI/NM_016734.3) from March 2015. The input 3′UTR length was 7285 nts, and the generated results defined by a score greater than 60 were an adequate prediction. Our last miRNA database was TargetScan (http://www.targetscan.org/vert_72/) (accessed on 20 March 2019) [47], a tool to search predicted microRNA targets in mammals that used a new version 7.2 released in March 2018 for miRNA prediction sites on a 7285 nts *Pax-5* 3′UTR.

## 3. Results

### 3.1. Pax-5 3′UTR Mapping and Polyadenylation Signal Prediction

Many genes undergo 3′UTR editing or truncation by way of APA signals to regulate gene expression [24,26,27]. Given the essential role of *Pax-5* in B-cell maturation and cancer processes, we set out to clone, sequence, and map *Pax-5* mRNA 3′UTR editing to gain a better understanding of post-transcriptional regulation leading to oncogenic *Pax-5* expression. Interestingly, the *Pax-5* transcript was characterized by a theoretical and approximated 7.2 kb sequence (NM_016734.3) [48]. To examine the *Pax-5* transcript for putative polyadenylation signals, we made use of a series of polyA signal prediction tools such as: DNA polyA signal miner [40], PolyApred [41], and Softberry POLYAH [42] to generate a list of predicted sites at their respective nucleotide positions in relation to the *Pax-5* stop codon (Appendix A). Canonical and non-canonical polyA signals identified from each database were then analyzed and compared. As expected, all polyA signals from the *Pax-5* 3′UTR were not predicted by every database because of the software’s respective algorithms and score thresholds. We identified up to 12 different and unique polyA signal sequences on the *Pax-5* 3′UTR where many canonical polyA consensus signals (5′-AATAAA-3′) were identified by multiple prediction databases (Appendix A). Interestingly, two canonical sites located at 4912 and 6019 nts beyond the stop codon (i.e., 3′UTR lengths of 4912 and 6019, respectively) were predicted by all polyA prediction tools used.

To validate our computational predictions, we performed 3′ rapid amplification of cDNA ends (3′ RACE) on a series of cell model types commonly known to express endogenous *Pax-5* to map *Pax-5* 3′UTR truncation generated by APA processes. Using B-cell lines (REH, Nalm6, and Raji) and breast epithelial cells (MCF7 and MCF10A), we cloned, sequenced, and aligned approximately 2000 amplified products from the *Pax-5* mRNA 3′UTR. Interestingly, we observed that the overall *Pax-5* 3′UTR length and structure varied extensively among cell types examined (Figure 1). Specifically, in mammary epithelial models, all *Pax-5* 3′UTRs from MCF7 cancer cells were truncated at 3.3 kb downstream of the stop codon, whereas 3′UTRs from MCF10A (non-cancerous) terminated at 3.9 kb (Figure 1A). These *Pax-5* 3′UTR regions from mammary cells thus represented a loss of approximately 3.9 kb (MCF7) and 3.3 kb (MCF10A) at the 3′ extremity of the 3′UTR in comparison to the reported 7.2 kb “full-length” 3′UTR of the *Pax-5* transcript (NCBI/NM_016734.3) [48]. We also found that the lengths and truncation of mammary cell-derived *Pax-5* 3′UTRs correlated with the predicted non-canonical polyadenylation signals located at approximately 3230 and 3902 nts, respectively, downstream of the stop codon.

On the other hand, *Pax-5* 3′UTR editing in B-cells was characterized by transcripts bearing 3′UTRs of various lengths and sequence profiles. In general, *Pax-5* 3′UTRs from B-cells did not exhibit premature truncation and preferentially used canonical polyA signals located at the distal 3′ end of the UTR. Interestingly, B-cell 3′UTRs underwent drastic splicing events (deletions up to 5.7 kb) resulting in a significant excision of the internal UTR regions within the *Pax-5* transcript (Figure 1B). To map the various spliced 3′UTR profiles of *Pax-5* transcripts in B-cells, we studied predicted splicing and regulatory RNA motifs using computational analyses (RNAreg 2.0) [43]. We found that sequenced regions from *Pax-5* 3′UTR splicing events correlated with predicted acceptor and donor regions of putative splicing sites (Figure 1B). These findings strongly suggested that *Pax-5* transcription utilizes APA signals and alternative splicing events in a tissue-specific manner to elicit *Pax-5* 3′UTR editing.

### 3.2. Pax-5 3′UTR Alternative Splicing in Primary B-Lymphocytes and Clinical Samples

Given the notable differences in *Pax-5* 3′UTR splicing events found in B-cell lines, we expanded our investigation to examine B-lymphocytes isolated from primary peripheral blood mononuclear cells (PBMCs) from healthy donors and patients suffering from B-cell cancer lesions (follicular lymphoma/FL, chronic lymphocytic leukemia/CLL, and diffuse large B-cell lymphoma/DLBCL). To map *Pax-5* 3′UTR alternative splicing in primary B-cells, we performed 3′RACE to target and amplify the flanking sequences of the 3′UTR regions undergoing splicing and excision. As expected, splicing within the *Pax-5* 3′UTR was observed in all samples tested (i.e., 6 healthy donors and 17 cancer patients) (Figure 2). Interestingly, clinical cancer samples demonstrated greater excised 3′UTR regions in comparison to healthy donors. Also, the spliced region within the 3′UTR of healthy donors occurred more distally of the stop codon (approximately 1 kb downstream), whereas splicing within the 3′UTR in cancer cells occurred more upstream and proximal (approximately 300 nts) of the stop codon (Figure 2). In addition, the majority of *Pax-5* 3′UTRs sequenced in B-cell samples correlated with polyA-predicted sites located at approximately 6 kb in contrast to mammary cells where *Pax-5* transcripts elicited early truncation by APA. These results supported our initial findings suggesting that *Pax-5* 3′UTR editing profiles appear to be tissue-dependent (mammary epithelial versus B-cells) and undergoing distinct molecular mechanisms (early truncation versus alternative splicing), respectively. Our observations also presented significant differences in polymorphic patterns of *Pax-5* 3′UTRs from clinical samples in comparison to healthy donors (i.e., greater excised 3′UTR regions) where cancer cells possessed overall shorter 3′UTRs.

### 3.3. UTR Editing of the Pax-5 Transcript Affects Ribosomal Translation

Given that oncogenes are known to shorten their 3′UTRs as a means to evade post-transcriptional regulation [27], we set out to assess the translational repercussions of 3′UTR editing of *Pax-5* transcripts. To do so, polysomal fractionation combined with PCR was performed on B-cell lines (REH and Raji) and primary B-cells to determine potential differences in translational frequency between *Pax-5* 3′UTR-edited regions. Polysomal fractions were thus collected into two collection clusters where the lighter fragments representing low ribosome-bound mRNA (low translation frequency) were separated from the heavy fractions characterized by polyribosomes and high translation frequency. Total RNA from these groups was then purified for RT-PCR amplification, cloning, and sequencing. As expected, the heavier polyribosome complexes were bound to *Pax-5* transcripts bearing shorter 3′UTRs in all B-cell lines tested (lanes 2 and 4, Figure 3). On the other hand, the fractions with low ribosomal complexes were associated with *Pax-5* mRNAs with longer 3′UTR regions (lanes 1 and 3, Figure 3). Equally, primary B-cells from healthy donors also displayed shorter 3′UTR lengths complexed with heavier polysomal complexes (lane 6, Figure 3). However, primary B-cells, which generally express longer *Pax-5* 3′UTRs in comparison to cancer cells (Figure 2), also demonstrated these features in both light and heavy polysomal complexes (lanes 5 and 6) when compared with cancer cells (lanes 1 to 4, Figure 3). These results suggested that B-cell *Pax-5* transcripts bearing shorter 3′UTRs are translated at a higher frequency in comparison to transcripts with longer 3′UTRs. Interestingly, when we repeated the polysomal fractionation assays with breast cancer cells, no differences were observed between polysomal density fractions. These observations may be a result of the relative minor differences in *Pax-5* 3′UTR editing in breast cell lines in terms of overall length and structure. Together, these results demonstrated that the overall length and structure of the *Pax-5* 3′UTR determine ribosomal occupancy leading to translation efficiency.

### 3.4. Impact of 3′UTR Editing on Pax-5 Post-Transcriptional Regulation

Given that *Pax-5* transcripts bearing shorter 3′UTRs were preferably complexed with heavy polysomal fractions and high translation frequency, we set out to validate our findings using *luciferase*-based reporter assays. Using the pMiR-Report construct containing the *luciferase* gene under the CMV promoter (Figure 4A), we cloned the most abundant 3′UTR regions from B-cell cancers into the multiple cloning site located at the 3′ end of the *luciferase* sequence. The first construct (i.e., pMiR-6kb) contained the longest identified continuous *Pax-5* 3′UTR sequence of the *Pax-5* 3′UTR region. Our second construct (pMiR-1kb) consisted of the 3′UTR with an approximate total length of 1 kb represented by upstream (1–284 nts) and downstream (4982–5821 nts) sequences of the 3′UTR. In other words, the pMiR-1kb contained a 3′UTR with an internally spliced region of +285 to +4981 nts relative to the stop codon. Our final construct (pMiR-0.3kb) consisted of the 3′UTR with a spliced region excising nucleotides +52 to +5619, resulting in an approximate remaining 0.3 kb 3′UTR length (Figure 4A).

Reporter constructs were then transiently transfected into B-cell cancer cell lines (Raji and REH) where luciferase signals were normalized on transfection efficiency. In REH cells, we observed that reporter constructs bearing shorter 3′UTRs (i.e., pMiR-1kb and pMiR-0.3kb) exhibited significantly greater relative luciferase activity (83 and 79%, respectively) in comparison to plasmids with longer 3′UTR regions (pMiR-6kb) (Figure 4B). Similarly, transfections of Raji B-cells with reporter genes characterized by shorter 3′UTRs (i.e., pMiR-0.3kb) exhibited greater luciferase activity compared to plasmids with longer 3′UTRs (i.e., pMiR-6kb and pMiR-1kb) (Figure 4C). Interestingly, we also observed slight differences between the REH (immature) and the Raji (mature) B-cell models in terms of relative reporter activity associated with the pMiR-1kb construct. We believe that these variations may be due to cell type, lineage maturity, post-transcriptional regulating elements, and differential miRNA profiles from each cell type. Globally, our results demonstrated that mRNA translation frequency is significantly regulated by the *Pax-5* 3′UTR length and structure (sequence). These observations further strengthened the role and impact of 3′UTR editing in aberrant and oncogenic *Pax-5* expression associated with B-cell cancers.

It has been well described that mRNA 3′UTR editing is a prominent process for oncogenes to avoid miRNA regulation [26,27]. To assess the impact of *Pax-5* 3′UTR shortening on its susceptibility to (or evasion of) miRNA regulation, we studied the activity of miRNA targeting on the internal excised regions found on shorter 3′UTRs from the *Pax-5* transcript. Predicted miRNAs targeting the spliced 3′UTR region (nucleotides +52 to +5619) were established by web-based tools and cross-referenced with miRNAs from the literature for B-cell expression levels and role in B-cell biology or oncogenesis (Appendix A). Three miRNAs were thus selected (i.e., miR-1275, 217-5p, and 181a-5p) and validated for their expression in our B-cell models using RT-qPCR (Figure 5A). Given that only REH cells concomitantly expressed all three selected miRNAs, these cells were co-transfected with a pMiR-Report construct (pMiR-6kb or pMiR-0.3kb) and an anti-miRNA (inhibiting each selected miRNA).

Although anti-miR inhibitors effectively suppressed the levels of each selected miRNA (miR-1275, miR-217, and miR-181a) over 80% (Figure 5B), we were unable to determine whether any single miRNA tested could independently alter significant changes in luciferase reporter activity from either long (pMiR-6kb) or shortened/spliced 3′UTRs (pMiR-0.3kb) (Figure 5C). In fact, converse to our expectation, the inhibition of miR-1275 and miR-181a decreased the average luciferase activity from the pMiR-6kb construct harboring binding sites for these selected miRNAs. These results may be due to indirect mechanisms or downstream targets affected by the silenced miRNAs. Also, given the significant size of the excised 3′UTR region (5.7 kb), this sequence may encompass a myriad of post-transcriptional regulating elements and binding motifs.

### 3.5. Shortening of Pax-5 3′UTR Is Prevalent in B-Cell Cancer Malignancies

Given the oncogenic potency of *Pax-5* products in B-cell malignancies, we hypothesized that aberrant overexpression of *Pax-5* in these cancers may be due to 3′UTR shortening. As shown previously, *Pax-5* 3′UTR length inversely correlated with its reporter translation frequency. To validate these findings, we made use of RNA-seq data sets (The Cancer Genome Atlas and Gene Expression Omnibus) to examine *Pax-5* 3′UTR editing in the most common B-cell malignancies characterized with aberrant *Pax-5* (ex: diffused large B-cell lymphoma/DLBCL [16,49] and chronic lymphocytic leukemia/CLL [50]) in comparison to healthy primary B-lymphocytes. We also analyzed 3′UTR editing in a series of common B-cell lines for a better conclusion. First, the expression data were compiled and plotted for *Pax-5* differential expression. Comparison analysis of B-cell lines versus primary B-cells indicated a 6.53-fold increase in *Pax-5* transcripts in this group (*p* < 0.001, Figure 6A). Accordingly, the overall expression level of *Pax-5* in DLBCL and CLL cells was also significantly higher than normal B-cells, with changes of about 9.14- and 8.58-fold, respectively (*p* < 0.001).

We next analyzed *Pax-5* 3′UTR shortening and its distribution within B-cell cancer subsets. Globally, transcript profiling of *Pax-5* 3′UTR lengths from B-cell cancers and B-cell lines demonstrated a significant shortening compared to healthy primary B-cells. For example, 75% of *Pax-5* 3′UTR sequences from primary B-cells extended beyond 1 kb in length, whereas only 13% of 3′UTRs were less than 200 nts (pie chart outer ring, Figure 6B). On the other hand, B-cell lines expressed 55% of 3′UTR sequences greater than 1 kb and 23% of 3′UTRs < 200 nts (inner ring, Figure 6B). This *Pax-5* 3′UTR landscape translates into B-cell cancer cell lines undergoing a 50% decrease (*p* < 0.001) in relatively long 3′UTRs (˃1 kb) and a 56% increase (*p* < 0.001) in short 3′UTRs (<200 nts) when compared to primary healthy B-cells.

Consistent with these results, analysis of DLBCL and CLL B-cells also expressed a significant shortening of *Pax-5* 3′UTR lengths. Data in CLL patients showed a 4.15-fold increase in *Pax-5* transcripts with short 3′UTR lengths (<200 nts) compared to control primary cells (*p* < 0.001, Figure 6C). Similarly, DLBCL expressed 4.84-fold more *Pax-5* transcripts with short 3′UTRs (<200 nts) compared with primary B-cells (*p* < 0.001, Figure 6D). Overall, these data indicated that the shortening of *Pax-5* 3′UTRs is widespread in B-cell cancer lesions.

### 3.6. Pax-5 3′UTR Shortening Relates with Hematopoietic Malignancy

We showed that cancer B-cells are characterized by a significant shortening of *Pax-5* 3′UTR regions in comparison to their healthy counterparts. We postulated that 3′UTR shortening may be associated with disease progression. To examine this, we measured overall lengths of *Pax-5* 3′UTRs by RT-PCR from patients with B-cell malignancies in relation to patient clinical staging of hematopoietic disease. Interestingly, the lengths from amplified *Pax-5* 3′UTRs demonstrated a trend with hematopoietic disease malignancy (Figure 6D). Specifically, we found that B-cells from healthy donors and B-cell cancers at early stages (1 and 2) appeared to contain longer *Pax-5* 3′UTR regions in comparison to patients with advanced disease (stage 3 and 4), which comprised significantly shorter 3′UTRs (Spearman’s rank correlation, r*_s_* = −0.8) (Figure 6D). These findings suggested that *Pax-5* 3′UTR shortening may correlate with B-cell hematopoietic disease progression. These results also brought physiological significance to 3′UTR editing in *Pax-5* aberrant expression and hematopoietic malignancy.

## 4. Discussion

It is well established that adequate and strict regulation of *Pax-5* expression is essential for B lymphocyte development and differentiation (reviewed in [51]). Conversely, aberrant *Pax-5* expression is involved in the pathogenesis of B-cell lymphoid cancer and, more recently, of solid carcinomas [8,17,52,53]. Although the downstream activity and phenotype associated with *Pax-5* expression are relatively well characterized, there is a paucity of studies elucidating the upstream regulatory mechanisms governing *Pax-5* expression in healthy and malignant cellular settings. In this study, we described the first evidence of *Pax-5* 3′UTR editing and its effects on translation in human B-cells and lymphoid cancer lesions. Specifically, we demonstrated that the *Pax-5* transcript utilizes alternative polyadenylation (APA) signals (both canonical and non-canonical) in addition to alternative splicing to alter its 3′UTR sequence and translational expression.

First, we profiled *Pax-5* transcripts in mammary cell lines and found that the majority of *Pax-5* 3′UTRs measured between 3.3 and 3.5 kb, which was seemingly shorter than the putative reported 7.2 kb full-length sequence (NCBI/NM_016734.3) [48]. Also, in mammary cell lines, termination of *Pax-5* transcripts consistently aligned with non-canonical polyA signals within the 3′UTR. Conversely, *Pax-5* 3′UTRs from B-lymphocytes depicted a much more diverse pattern of mRNA editing consisting of alternative splicing events and the use of canonical polyA signals. These findings thus described marked differences in *Pax-5* 3′UTR editing and in post-transcriptional regulation, which appeared to be tissue-dependent. Accordingly, these discrepancies in *Pax-5* 3′UTR editing were consistent with *Pax-5* function, which was shown to exhibit dichotomous phenotypes depending on its cellular context. For example, *Pax-5* promoted an oncogenic role in specific lymphoproliferative cancers (ex: CLL and DLBCL), whereas in mammary cancers, *Pax-5* elicited tumor suppressor features and promoted cellular epithelialization [53,54].

We also observed that *Pax-5* mRNAs from B-lymphocytes primarily aligned with distal canonical polyA sites (compared to proximal sites used by mammary cells). More importantly, B-cell *Pax-5* transcripts were subjected to splicing and excision of internal regions in the 3′UTR (up to 5.7 kb). To our knowledge, this was the first evidence of 3′UTR alternative splicing of *Pax-5* transcripts. These findings were envisaged given multiple reports of *Pax-5* transcriptional editing processes at both the 5′ and 3′ extremities [10,13,55]. In fact, studies have estimated that 95% of gene transcripts undergo alternative splicing events [1,2,3] where only 4% of these events occur in the 3′UTR [56]. Furthermore, mRNAs encoding transcription factors are more commonly spliced (accounting for 14%) than other genes [57]. Although mRNA editing is vital for the expansion of gene products and function diversity, aberrant alternative splicing is also concomitantly linked to cancer [4]. Accordingly, our observation of extensive editing of *Pax-5* 3′UTRs in cancer cells may explain aberrant and oncogenic *Pax-5* expression. This hypothesis was further supported by the association of *Pax-5* 3′UTR shortening to B-cell cancer progression.

Our findings concurred with previous reports indicating that shortening of mRNA 3′UTRs has major repercussions on translation, stability, and function [25,58]. Some studies have reported that 3′UTR shortening is a prominent event primarily observed in proliferating cells and oncogenes [26,27]. Our results supported these findings where *Pax-5* overexpression in B-cell cancer cell lines and patient samples (i.e., CLL and DLBCL) are marked with significant shortening of *Pax-5* 3′UTRs. Most importantly, we associated 3′UTR shortening to various clinical stages of B-cell cancer progression. We thus propose that aberrant *Pax-5* expression in B-cell cancer arises from 3′UTR editing to escape post-transcriptional regulation from co-interacting elements. Considering the extent of the 3′UTR editing of the *Pax-5* transcript (5.7 kb deletion), we initially focused on *trans*-acting regulators such as miRNAs. It is well established that miRNAs convey an effective level of gene regulation leading to altered expression profiles and cancer cell behavior [59]. Mechanistically, miRNAs inhibit translation through the specific targeting of complementary sequences located on the target mRNA, notably in the 3′UTR [60]. We thus generated a list of predicted miRNAs targeting the *Pax-5* 3′UTR in cancer cells (Appendix A). As a result, the *Pax-5* transcripts from breast cancer cells characterized by 3′UTRs 3.3 kb would theoretically escape more than half (51.5%) of the targeting miRNAs in comparison to the reported full-length 3′UTRs. More interestingly, *Pax-5* transcripts from B-cells characterized by a 5.7 kb excision of the 3′UTR internal sequence would evade approximately 94% of the predicted targeting miRNAs. *Pax-5* 3′UTR editing is thus a determining factor of miRNA-mediated regulation of the *Pax-5* oncogene in cancer cells.

Many of the miRNAs identified in Appendix A are uncharacterized while others, such as miR-181a, miR-217, and miR-1275, have been reported as tumor suppressors or involved in cancer disease progression [61,62,63,64]. Suitably, we attempted to study the role of miR-1275, miR-217, and miR-181a in post-transcriptional regulation of *Pax-5,* given that these miRNAs and cognate binding motifs would be specifically lost following the editing and deletion of 3′UTR internal sequences. Although the overall length and structure (sequence) of *Pax-5* 3′UTRs had a significant effect on reporter activity (Figure 4), the inhibition of each selected miRNA did not alter the expression of reporter genes harboring long (pMiR-6kb) or short (pMiR-0.3kb) 3′UTRs (Figure 5). The latter tested miRNAs may be unable to solely regulate 3′UTR-dependant *Pax-5* expression owing to the considerable size of the spliced region (5.7 kb) and its potential interactions with other post-transcriptional regulating elements. As mentioned previously, the 5.7 kb spliced internal region of the *Pax-5* 3′UTR would evade approximately 94% of all predicted targeting miRNAs (Appendix A).

In addition to miRNA regulation, adenylate/uridylate-rich elements (AU-rich elements/AREs) are also prevalent mRNA stability motifs located in 3′UTRs. These motifs allow RNA-binding proteins (RBPs) to regulate translation by targeting ARE-containing mRNAs for rapid degradation (reviewed in [65]). These *cis*-acting regulatory elements within 3′UTRs are most commonly found in human genes encoding proto-oncogenes, transcription factors, and cytokines involved vital cellular processes (ex: stress response, cell-cycle regulation, inflammation, apoptosis, etc.) and carcinogenesis [44,66]. The accessibility and interplay between AU-rich regulatory motifs and their respective RBPs within *Pax-5* 3′UTR variants may also contribute to the fate of mRNA stability and translation. Using AU-rich motif prediction software (RegRNA 2.0 [43] and ARED-Plus [44]), we identified four ARE sites (defined by a core AUUUA sequence motif) at positions +3767, +5702, +7091 and +7137 within the *Pax-5* 3′UTR. In this case, *Pax-5* transcripts from B-cells with alternatively spliced 3′UTRs would be devoid of 75% of AREs, resulting in increased *Pax-5* translation. Altogether, the truncation and excision of important regulatory motifs within the *Pax-5* 3′UTR will curtail *cis*- and *trans*-acting elements in an orchestrated manner to regulate mRNA half-life and protein translation efficacy [60,66,67].

Overall, our findings provide new evidence that the human *Pax-5* transcript is submitted to 3′UTR editing (APA and alternative splicing). Shorter *Pax-5* 3′UTRs also promote downstream expression as demonstrated by increases in reporter gene activities. In addition, we found that *Pax-5* 3′UTR lengths are associated with B-cell cancer lesions and malignancy. Our study not only brings new insight to the molecular biology of the post-transcription processing of *Pax-5* mRNA but also provides new potential mechanisms for aberrant *Pax-5* expression in cancer cells. Globally, these preliminary findings warrant the need for further investigation as potential therapeutic options or diagnostic tools.

## Figures and Tables

**Figure 1 cells-11-00076-f001:**
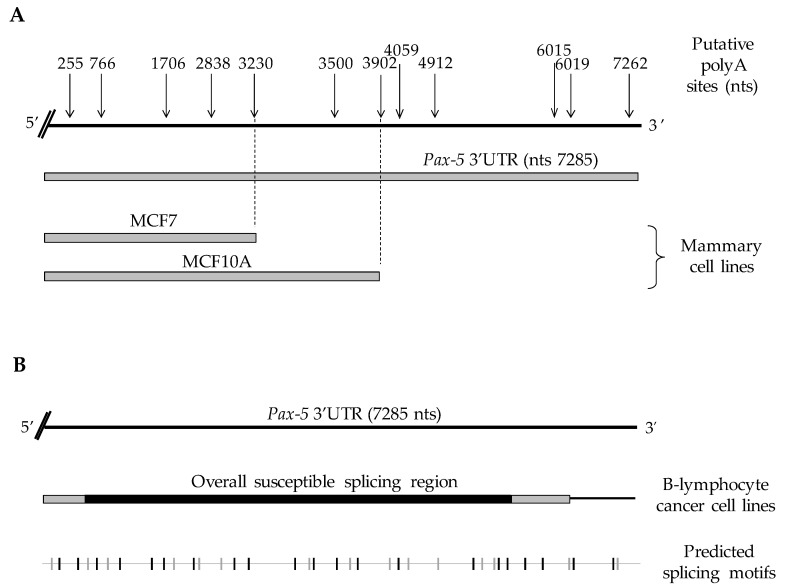
Schematic representation of *Pax-5* 3′UTR editing in various cell types. *Pax-5* 3′UTR sequences isolated from (**A**) breast and (**B**) B-cell lines (Raji, REH and Nalm6) were amplified using 3′RACE, sequenced and aligned with the genomic *Pax-5* 3′UTR (top strand) marked with the predicted alternative polyA sites (top arrows) in relation to their nucleotide position after the stop codon. The bottom line represents a computational analysis of *Pax-5* 3′UTR putative splicing elements where the gray vertical nicks represent acceptor sites, whereas black vertical nicks represent donor elements for splicing sites (http://regrna2.mbc.nctu.edu.tw) (accessed on 20 March 2019).

**Figure 2 cells-11-00076-f002:**
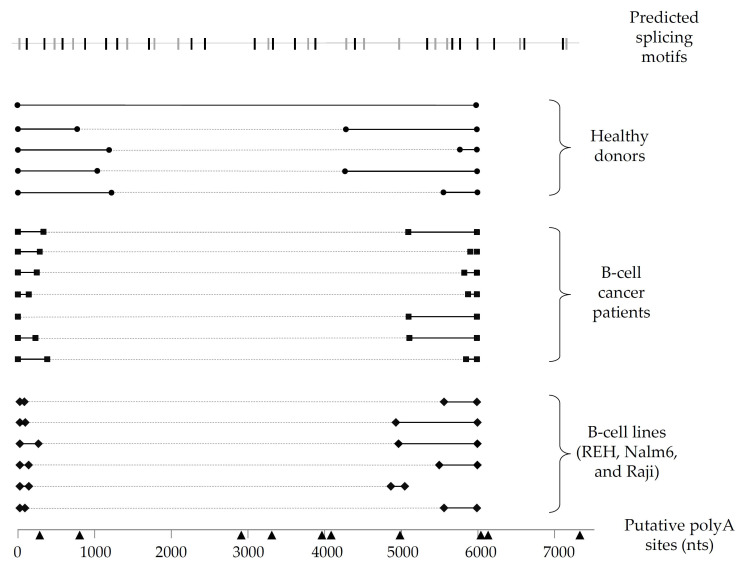
Schematic representation of *Pax-5* 3′UTR editing in B-lymphocytes. *Pax-5* 3′UTR profiling of splicing patterns was performed in B-lymphocytes from healthy donors (filled circles), B-cell cancer patients (filled squares), and B-cell cancer lines (filled diamonds). The bold black lines represent the complementary alignment with the referenced full-length *Pax-5* 3′UTR sequence (NCBI/NM_016734.3) [48], whereas the dotted gray lines represent the excised sequences from the spliced region. Putative polyA signals (black triangles) located at their respective nucleotide position (bottom line) were predicted for polyA canonical and non-canonical motifs using bioinformatics software tools (i.e., Softberry POLYAH, DNA polyA signal miner, and PolyA pred).

**Figure 3 cells-11-00076-f003:**
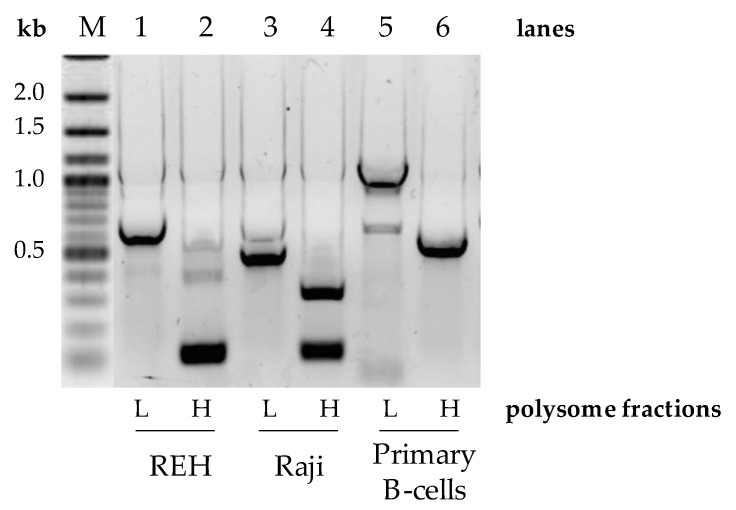
*Pax-5* 3′UTR profiling from polysomal fractionation. The mRNA from polysomal fractions of B-cell populations was extracted, amplified by RT-PCR, and sequenced. Polysomal light (L) fractions represent low translational activities (none or few ribosomes attached to the mRNA), whereas heavy (H) fractions represent active mRNA translational complexes. PCR primers were designed to flank the excised spliced regions of the *Pax-5* 3′UTR and be compared between B-cell lines (REH and Raji) and primary CD19+ B-lymphocytes from healthy donor PBMCs. “M” indicates the reference molecular weight marker with corresponding nucleotide length kilobases (kb).

**Figure 4 cells-11-00076-f004:**
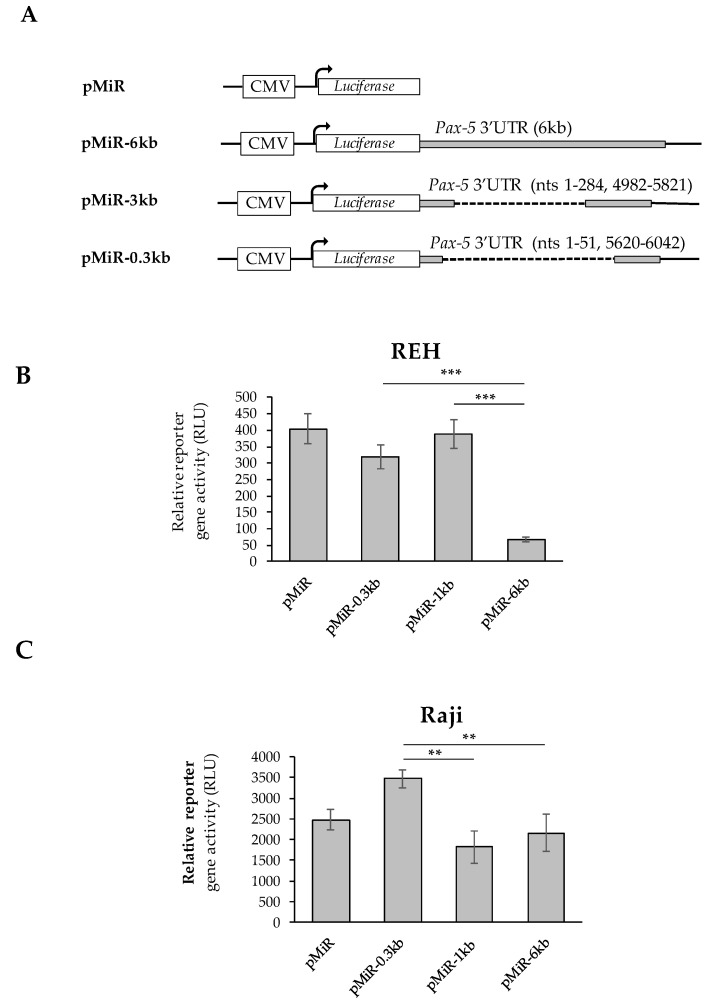
Effects of *Pax-5* 3′UTR editing on translation potential. The most prominent spliced 3′UTR isoforms from *Pax-5* transcripts isolated from B-cells were (**A**) cloned downstream of the firefly luciferase reporter gene. The pMiR-REPORT constructs were either left empty (pMiR) or contained a full-length *Pax-5* 3′UTR (pMiR-6kb); a 3′UTR variant of 1 kb (pMiR-1kb) or a 3′UTR variant of 0.3 kb (pMiR-0.3kb). The respective lengths and structures from each 3′UTR isoform and plasmid are indicated with their respective nucleotide (nt) positions relative to the stop codon. (**B**) REH and (**C**) Raji B-cell lines were transiently transfected with pMiR reporter constructs and evaluated for relative firefly luciferase activity. Relative light units (RLUs) were normalized for each transfection condition using qRT-PCR on puromycin (a resistance gene located on the backbone of the reporter plasmid). The presented data are the calculated means of three independent samples, where statistical analysis by *t*-test indicates significant differences with respect to control cells (** *p* < 0.01 and *** *p* < 0.001).

**Figure 5 cells-11-00076-f005:**
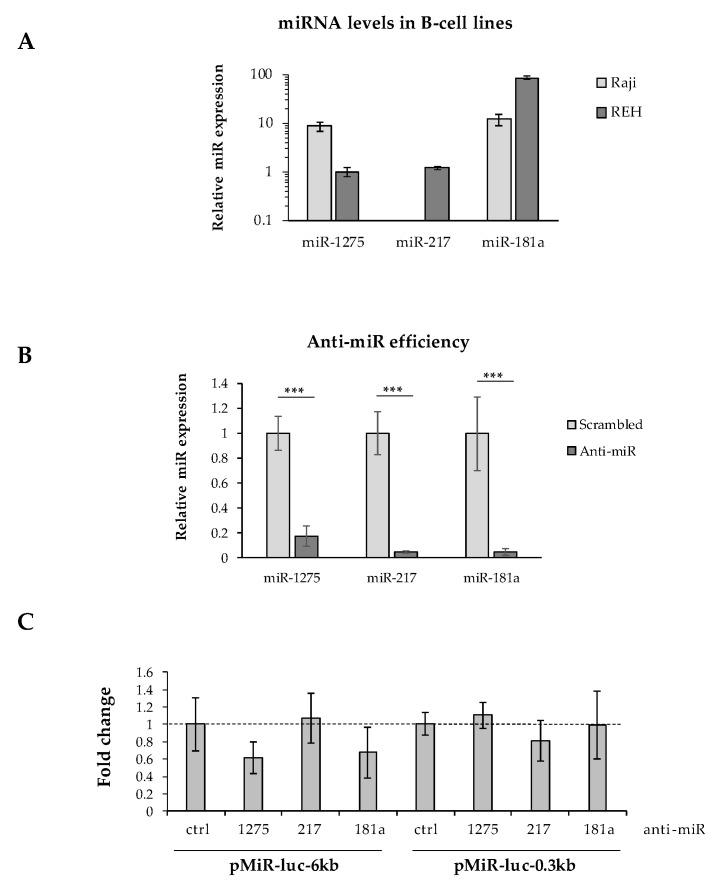
Effects of *Pax-5* 3′UTR shortening on miRNA regulation. (**A**) Relative expression levels of miR-1275, miR-217, and miR-181a were determined in Raji and REH B-cell lines using TaqMan qPCR. Expressed levels were normalized and plotted in relative fold log expression over control samples (RNU48 expression). (**B**) REH cells were transiently transfected with anti-microRNA (anti-miR) inhibitors against miR-1275, miR-217, and miR-181a for 48 h. MiRNA silencing was then evaluated by TaqMan qPCR and compared to their respective transfected non-targeting controls (scrambled). MiRNA levels were normalized and plotted in relative expression over control samples (scrambled anti-miRs). (**C**) REH cells were transiently co-transfected with pMiR reporter constructs (pMiR-6kb versus pMiR-0.3kb) and anti-miR inhibitors (1275, 217, and 181a) then evaluated for relative luciferase activity. Relative light units (RLUs) were normalized and plotted in relation to control samples (scrambled anti-miRs) represented by a dotted line. The presented data are the calculated means of three independent biological assays, where statistical analysis by *t*-test indicates significant differences in respect to controls (*** *p* < 0.001).

**Figure 6 cells-11-00076-f006:**
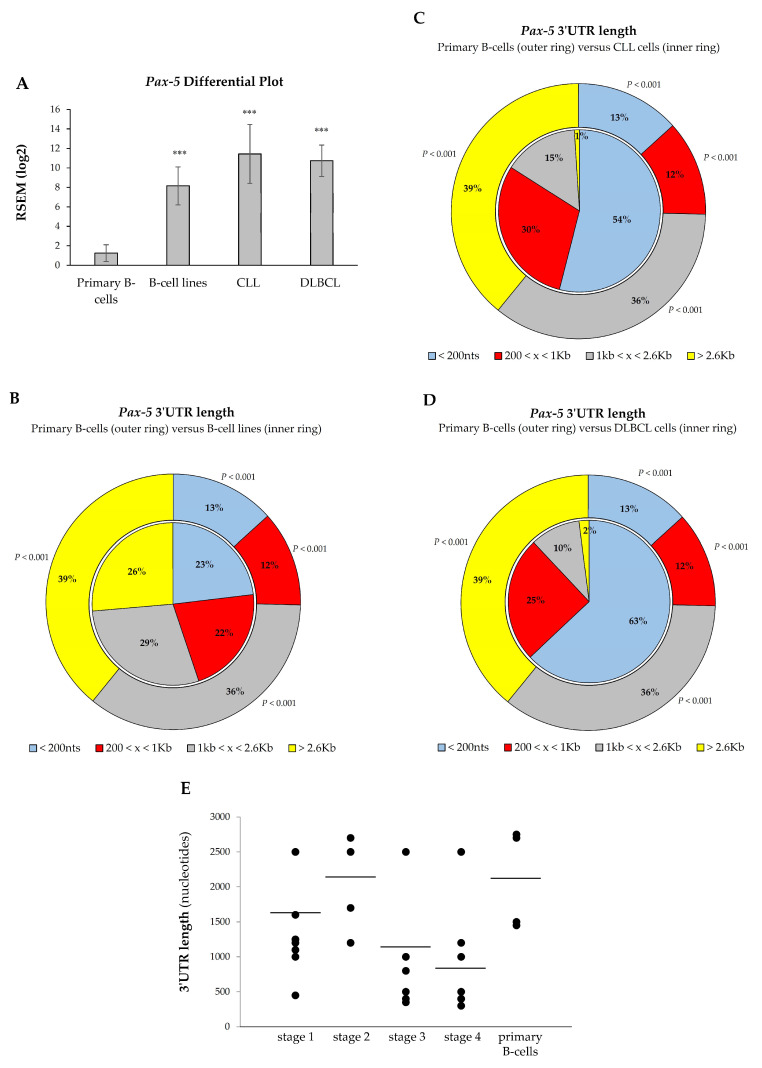
3′UTR shortening profiles in B-cell cancer lesions. Comparison analyses of (**A**) *Pax-5* transcript levels and overall 3′UTR length distribution were determined in (**B**) immortalized B-cell lines, (**C**) chronic lymphocytic leukemia/CLL, and (**D**) diffused large B-cell lymphoma/DLBCL. Compilation of RNA-sequencing data for *Pax-5* was obtained from GSE16921 immortal B-cell data set (*n* = 41), GSE66117 CLL data set (*n* = 47), and TCGA-DLBCL data set (*n* = 48) versus two normal B-cell data sets GSE62246 (*n* = 2) and GSE70830 (*n* = 5), which were used as control samples (outer ring of panels B, C, and D). The read counts of these data sets were estimated by RSEM package (RNA-seq by expectation maximization). The asterisk and text indicate statistical differences by *t*-test analyses with respect to normal B-cell control samples (*** *p* < 0.001). (**E**) The *Pax-5* 3′UTR length was determined by RT-PCR in primary B-lymphocytes isolated from healthy donors (*n* = 7) and patients (*n* = 17) suffering from B-cell cancer lesions (Appendix A). The lengths of PCR-amplified products were then measured and plotted according to patients’ clinical disease staging. The average 3′UTR lengths in each category are indicated (black bars).

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
