# Peer review of "Pax-5 Protein Expression Is Regulated by Transcriptional 3′UTR Editing"

_cells, 2021, doi:10.3390/cells11010076_

Round 1
Reviewer 1 Report
This manuscript entitled “Pax-5 protein expression is regulated by transcriptional 3'UTR editing” is observational as the authors showed the experiments to validate the observation. This manuscript discusses the post-transcriptional regulation of Pax-5 expression and how Pax-5 3’UTR shortening correlates with increased translational occupancy in cancer. Additionally, the authors discovered the regulatory elements in 3’ UTR of Pax-5 which is regulated by microRNAs. This manuscript may encourage the researchers to develop precise molecular candidate for treatment. Below are my comments:
Major comments:
- In Raji B-cell line in pMiR transfected condition, some inhibitory effect is seen even though that does not have any regulatory sequences of Pax-5 3’UTR in downstream of the luciferase gene. However, pMiR 0.3Kb has a stimulatory effect. Please justify.
- Authors should justify for the same level of expression in the control vector (pMiR) and pMiR-1kb in REH B-cell line. The authors need to repeat this experiment with proper standardization in both B-cell cancer cell lines.
- Authors should show luciferase expression in other cell types to confirm these regulatory effects are only due to miRNA. Additionally, authors should show at least one other cell type in which there are no expressions of all three miRNAs and then perform the reporter gene assay.
- In figure 6A authors showed the transcript levels in primary B-cells, B-cell lines, CLL, and DLBCL. It would be useful to retrieve the proteomics data and analyze the abundance of Pax-5 proteins.
- If translation efficiency is regulated by shortening of Pax-5 3’UTR then why are the transcript levels varying across these cell lines (primary B-cells, B-cell lines, CLL and DLBCL).
- For polysome fractionation combined with PCR study authors should incorporate B-cell healthy cell lines for better control along with B-cell cancer cell lines.
- All bar graphs should be replaced with graphs that explicitly show at least 3 or more data points with mean and SD values.
- The author should make a Venn diagram for predicted miRNAs and also give the reasons for selecting only three miRNAs for validation.
- The authors should provide a schematic diagram to show the all primers (used in the entire study) and selected miRNA location in Pax-5 3’UTR region since the table is very hard to understand.
- In introduction, author should mention about Pax-5 5’UTR regulations.
Minor Comments:
- Please show the polysome fractionation assay profiling graph of both breast cancer cell line and B cell cancer cell lines with control in the supplementary figure.
- Page No.3, Line 103-112, there is a discrepancy in the use of fonts and size.
- Page No.3, Line 129, please provide the details of the nucleofection setting used for different cell lines.
- Page No.4, Line 141, please provide the percentage range of sucrose gradient used for the study.
- The authors should explain why they want to validate adenylate uridylate-rich elements in the method section.
- Page No.5, Line 196, authors should use “nts” throughout the manuscript in place of “nt”.
- Page No.8, Line 308, authors should use “ribosomal occupancy” in place of “ribosomal processing”.
- The authors should mention the gender of the healthy donor in supplementary table 4.
- In figure 4, authors should write “pMiR 0.3kb” instead of “pMiR 0,3kb”
Author Response
"Please see the attachment."

Reviewer 2 Report
In this study Beauregard et al., characterize the Pax-5 3’UTR region and its putative effects on Pax-5 mRNA splicing and regulation. Interestingly the study identifies that Pax-5 3’UTR editing events are highly prevalent in healthy B-lymphocytes with significant shortening an increased translation frequency in mature malignant B-cell cell lines.
The study provides novel evidence regarding Pax-5 regulation that is important for understanding oncogenesis. However, there are few concerns that need to be addressed.
- Throughout the manuscript the authors refer to different variants within Pax-5 as polymorphisms. Although many of the variants would correspond to true polymorphisms others (particularly in B-cell tumors) might be in fact nucleotide variants induced by somatic mutations. This issue needs to be clarified.
- In section 3.6, the manuscript claims that “Pax-5 3’UTR shortening correlates with hematopoietic malignancy”. From the provided figures seems that shorted reads are more prevalent in malignant B-cells than in primary B-cells, thus there seems to be an association between these two variables. Next a univariate analysis on disease stage Pax-5 3’UTR read lengths if provided. However, the patient cohort is extremely heterogeneous with 7 different diagnoses in 17 cases, moreover CLL stages do not necessarily correspond to DLBCL, FL or HL stages. There is no control for confounding variables or other earlier stablished disease progression factors. I suggest either to rephrase the findings to match the provided evidence, or to conduct a more robust correlation analysis between disease progression and Pax-5 3’UTR shortening including appropriate controls.
- This work describes regulatory consequences of nucleotide variants and the shortening of Pax-5 3’UTR in B-cell neoplasms. Do these findings translate into protein expression changes? In this regard the statement (lines 543-549): “Overall, our findings provide new evidence that the human Pax-5 transcript is submitted to 3’UTR editing (APA and alternative splicing), which modulates its translation efficacy.” Does not seem to be fully supported by the provided evidence (experiments showing changes in translation efficacy are no provided). If the data is not available this limitation should be clearly stated.
Minor issues:
- The dotted gray line in Figure 2 is very hard to read.
Author Response
"Please see the attachment."

Round 2
Reviewer 1 Report
Comments have been addressed and authors have made necessary modifications needed in the manuscript. However, I have the following concerns:
Minor comments:
-
It seems ribosomal fractionation assays graph provided in the
Supplementary Material are identical of same sample. Please clarify.
Author Response
"Please see the attachment."
